# Reduced Biliverdin Reductase-A Expression in Visceral Adipose Tissue is Associated with Adipocyte Dysfunction and NAFLD in Human Obesity

**DOI:** 10.3390/ijms21239091

**Published:** 2020-11-29

**Authors:** Valentina Ceccarelli, Ilaria Barchetta, Flavia Agata Cimini, Laura Bertoccini, Caterina Chiappetta, Danila Capoccia, Raffaella Carletti, Claudio Di Cristofano, Gianfranco Silecchia, Mario Fontana, Frida Leonetti, Andrea Lenzi, Marco Giorgio Baroni, Eugenio Barone, Maria Gisella Cavallo

**Affiliations:** 1Department of Experimental Medicine, Sapienza University of Rome, 00161 Rome, Italy; valentina.ceccarelli1@gmail.com (V.C.); ilaria.barchetta@uniroma1.it (I.B.); flaviaagata.cimini@uniroma1.it (F.A.C.); laura.bertoccini@uniroma1.it (L.B.); andrea.lenzi@uniroma1.it (A.L.); 2Department of Medical-Surgical Sciences and Bio-Technologies, Sapienza University of Rome, 04100 Latina, Italy; caterina.chiappetta@uniroma1.it (C.C.); danila.capoccia@uniroma1.it (D.C.); raffaella.carletti@uniroma1.it (R.C.); claudio.dicristofano@uniroma1.it (C.D.C.); gianfranco.silecchia@uniroma1.it (G.S.); frida.leonetti@uniroma1.it (F.L.); 3Department of Biochemical Sciences “A. Rossi-Fanelli” Sapienza University of Rome, 00185 Rome, Italy; mario.fontana@uniroma1.it; 4Department of Clinical Medicine, Public Health, Life and Environmental Sciences (MeSVA), University of L’Aquila, 67100 Coppito, Italy; marco.baroni@uniroma1.it; 5Neuroendocrinology and Metabolic Diseases, IRCCS Neuromed, 86077 Pozzilli, Italy

**Keywords:** biliverdin reductase-A, obesity, adipose tissue dysfunction, NAFLD, metabolic disorders

## Abstract

Biliverdin reductase A (BVR-A) is an enzyme involved in the regulation of insulin signalling. Knockout (KO) mice for hepatic BVR-A, on a high-fat diet, develop more severe glucose impairment and hepato-steatosis than the wild type, whereas loss of adipocyte BVR-A is associated with increased visceral adipose tissue (VAT) inflammation and adipocyte size. However, BVR-A expression in human VAT has not been investigated. We evaluated BVR-A mRNA expression levels by real-time PCR in the intra-operative omental biopsy of 38 obese subjects and investigated the association with metabolic impairment, VAT dysfunction, and biopsy-proven non-alcoholic fatty liver disease (NAFLD). Individuals with lower VAT BVR-A mRNA levels had significantly greater VAT IL-8 and Caspase 3 expression than those with higher BVR-A. Lower VAT BVR-A mRNA levels were associated with an increased adipocytes’ size. An association between lower VAT BVR-A expression and higher plasma gamma-glutamyl transpeptidase was also observed. Reduced VAT BVR-A was associated with NAFLD with an odds ratio of 1.38 (95% confidence interval: 1.02–1.9; χ^2^ test) and with AUROC = 0.89 (*p* = 0.002, 95% CI = 0.76–1.0). In conclusion, reduced BVR-A expression in omental adipose tissue is associated with VAT dysfunction and NAFLD, suggesting a possible involvement of BVR-A in the regulation of VAT homeostasis in presence of obesity.

## 1. Introduction

The prevalence of obesity has rapidly increased worldwide in the last decades and nowadays obesity represents a global health problem [1,2], contributing to the development of cardiovascular disease, type 2 diabetes mellitus (T2D), non-alcoholic fatty liver disease (NAFLD), dyslipidemia, chronic inflammation and cancer [3,4]. In response to chronic overnutrition and energy unbalance, adipose tissue (AT) undergoes dynamic remodelling with changes in adipocyte number and size, local hypoxia, immune cells infiltration, and aberrant fibrogenesis [5,6,7,8,9,10], leading to dysregulated secretion of AT-derived cytokines, hormones, and metabolites [5,6,10,11,12,13,14,15,16].

Biliverdin reductase A (BVR-A) is an enzyme with pleiotropic functions, mainly known for its role in heme metabolism, where it reduces biliverdin to bilirubin [17,18,19,20,21], an important antioxidant compound, contributing to protecting cells from oxidative stress [18,22,23,24]. During the last years, BVR-A emerged as a novel mediator/regulator of the insulin/IGF-1/PI3K/MAPK signalling cascade, and several studies have shown its crucial role in insulin-glucose homeostasis and glucose uptake [25,26,27,28,29,30,31]. BVR-A has also been shown to prevent liver lipid accumulation [32]. In particular, experimental mouse models knocked-out for hepatic BVR-A, on a high-fat diet (HFD), developed insulin signalling dysregulation and more severe glucose impairment and liver steatosis than wild type [32]. Moreover, the loss of adipocyte BVR-A has been associated with an altered metabolic and inflammatory profile in murine models of obesity [33]; AT-specific BVR-A knock-out mice exhibited increased fasting blood glucose levels and greater visceral fat accumulation in comparison to the wild type. Furthermore, loss of adipocyte BVR-A reduced mitochondria number and increased local inflammation and adipocyte size [33]. 

Recently, we have demonstrated that obese individuals have significantly lower BVR-A levels in peripheral blood mononuclear cells (PBMC) than lean matched controls and that reduced BVR-A levels are associated with an aberrant activation of the insulin signalling pathway, metabolic syndrome (MS) diagnosis, NAFLD and visceral adipose tissue (VAT) inflammation [34]. 

Although the overall evidence suggests a potential role of BVR-A in AT homeostasis, so far, no data are available on BVR-A expression in VAT in humans. Therefore, this study aimed to evaluate the expression of BVR-A in VAT of obese subjects and to investigate the association with markers of metabolic impairment, VAT dysfunction, and presence of NAFLD.

## 2. Results

Within our study participants, 60.5% fulfilled the criteria for MS diagnosis, 10.5% had T2D and 76.5% meet the criteria for NAFLD diagnosis. In our study population, the median value of BVR-A mRNA expression was calculated and the population study was stratified into two sub-groups, below and above the median value (low BVR-A group and high BVR-A group, respectively) (Figure 1). 

The two subgroups did not differ significantly in most parameters, except for GGT activities and NAFLD prevalence, which were significantly higher in subjects with low vs. high BVR-A levels. Characteristics of study participants in relation to VAT BVR-A mRNA expression are described in Table 1.

Patients belonging to the low BVR-A group had significantly increased VAT IL-8 and Caspase 3 mRNA expression than individuals with the high BVR-A group (*p* = 0.014 and *p* = 0.011, respectively; Figure 2A). Furthermore, individuals belonging to the low BVR-A group showed a trend with increased VAT IL-6 mRNA expression levels with respect to the individuals in the high BVR-A group, while no significant changes or associations were found for TNF-α levels (Figure 2B).

The evaluation of adipocytes’ size revealed a significant negative correlation between VAT BVR-A mRNA expression levels and adipocytes’ size (r = −0.72, *p* = 0.008) in our obese population, meaning that the lower BVR-A mRNA expression levels, the higher the size of the adipocytes. Indeed, individuals in the low BVR-A group have an increased adipocytes size on average (~20% larger), although this difference did not reach a statistical significance (Figure 3).

Furthermore, individuals belonging to the low BVR-A group had significantly higher GGT activities (*p* = 0.05) and prevalence of NAFLD (95% vs to 68%, *p* = 0.036) than those belonging to the high BVR-A group.

Of note, in a subgroup of individuals (*n* = 6) where BVR-A mRNA was not detectable in VAT, liver steatosis grade was significantly greater than what was reported in individuals with higher VAT BVR-A levels (*p* = 0.03).

Finally, individuals belonging to the low BVR-A group had an odds ratio for NAFLD of 1.38 (95% confidence interval:1.02–1.9; χ^2^ test); reduced VAT BVR-A predicted NAFLD with AUC = 0.89 (*p* = 0.002, 95% CI = 0.76–1.0) in ROC curve adjusted for age, sex and BMI (Figure 4).

## 3. Discussion

This is the first study investigating in humans the association between BVR-A mRNA expression in relation to VAT dysfunction and the presence of NAFLD. The main finding of our study is represented by the observation that obese individuals with lower VAT BVR-A mRNA levels display marked VAT impairment at the omental biopsy, as indicated by greater IL-8 and Caspase 3 mRNA expression levels than those reported in individuals with higher VAT BVR-A mRNA levels. Moreover, our data make an advancement in the comprehension of the mechanisms responsible for increased VAT inflammation in obese subjects, by confirming also in humans from previous observations collected in a murine model of obesity [33]. Indeed, Stec et al. reported that loss of BVR-A in mouse adipocytes is associated with increased visceral fat accumulation, greater adipocyte size, and higher local inflammation [33].

Results collected in the current study further strengthen the role of IL-8 and Caspase-3 in VAT dysfunction, by providing a potential mechanism associated with BVR-A and leading to their elevation. Indeed, as previously reported, BVR-A silencing leads to a significant increase of Caspase-3 levels in different cell types [35] whereas biliverdin administration significantly decreased the LPS-mediated IL-8 gene expression and secretion [36] thus supporting a role for BVR-A in controlling the inflammatory response [37]. In particular, the role of IL-8 and Caspase-3 in VAT inflammation was highlighted by several studies. Kobashi et al. have demonstrated that human adipocytes express the main receptor for IL-8 and that circulating IL-8 plays an autocrine effect on these cells by enhancing IL-8 mRNA expression [38], thus creating a vicious circle capable of starting and maintaining the inflammatory processes in the AT. In addition, it was demonstrated that IL-8 can promote macrophages infiltration in AT [39], influencing local and systemic inflammation and representing a possible link between AT impairment and insulin resistance-related diseases [38,39,40,41,42,43,44]. Moreover, Tinahones et al. have recently demonstrated that morbidly obese individuals exhibited increased AT CASP3/7 expression along with reduced BCL2 expression, and this proapoptotic state correlated with an increased gene expression of inflammatory cytokines and with alteration of insulin signalling [45]. Taken together, these data suggest a possible relationship between inflammatory processes, apoptotic pathways, and insulin signalling impairment, which characterize obese subjects and may explain the increased susceptibility of these individuals to developing insulin resistance.

In our study, no significant differences or association were found for IL-6 and TNF-α expression between low and high BVR-A group, probably because IL-6 and TNF-α were released in very small amounts by adipocytes and the release of these interleukins is mainly due to the non-fat cells present in human AT [46]. Another possible reason is that IL-6 and TNF-α are cytokines that undergo very small variations and therefore a larger study population is needed to reach statistical significance. Our study is properly powered for demonstrating differences in VAT inflammation, i.e., IL-8 expression in VAT, which we think may be the cytokine able to detect better VAT dysfunction [15,16].

Another intriguing finding of our study relies on the observation that obese individuals in the low BVR-A group display larger adipocytes size than obese individuals in the high BVR-A group. In the last years, several studies have focused on the relationship between adipocyte size and inflammation [47,48] as well as impaired glucose-insulin homeostasis and diabetes [49,50,51], by demonstrating that increased adipocyte size is associated with a metabolically unfavorable profile in obese individuals. Furthermore, our data support a role for BVR-A in regulating adipocyte size also in human obesity in agreement with the observation by Stec et al. showing that AT-specific BVR-A knock-out (KO) mice exhibited greater adipocyte size as compared to wild-type [33].

In addition, BVR-A, other than the role in the regulation of insulin signaling pathway as mentioned above [25,26,27,28,29,30], is historically known because is the enzyme responsible for bilirubin production [17,18,19,20,21]. Clinical evidence suggests that individuals with mildly elevated bilirubin levels have significantly fewer metabolic disorders such as fatty liver, obesity, or T2D [52,53,54,55,56]. Increased bilirubin serum levels were shown to decrease body weight and to reduce lipid accumulation in experimental models [57,58], likely by a direct interaction between bilirubin and the peroxisome proliferator-activated receptor α (PPARα), resulting in the activation of lipid-burning genes -such as UCP1 and CPT1- in VAT adipocytes [59]. Bilirubin-mediated PPARα-induced activation of UCP1 and CPT1 also results in enhanced mitochondrial function, finally promoting lipid burning and weight loss [59]. Altogether, these findings suggest that bilirubin may regulate VAT tissue expansion, reduce VAT hypertrophy, and improve glucose metabolism. We did not observe differences in bilirubin levels between obese individuals with low and high BVR-A mRNA levels. This observation is not surprising to us, because we evaluated BVR-A mRNA expression levels in VAT, while bilirubin metabolism is a central mechanism in the liver, necessary to normal hepatocyte function [17,19]. Hence, changes in VAT BVR-A mRNA levels are unlikely to explain changes in circulating bilirubin levels. Rather, our results on smaller adipocytes size in the high-BVR-A group are in line with the above-cited observations and support the hypothesis that higher BVR-A levels could favour an increased bilirubin production within the adipocytes thus promoting lipid-burning effects.

Our work shows also that lower VAT BVR-A mRNA expression is associated with histological diagnosis and severity of NAFLD among obese individuals. This is in line with a recent study showing that liver-specific BVR-A KO mice had significantly increased hepatic fat accumulation than wild type [32], thus suggesting that loss of BVR-A results in exacerbation of hepatic steatosis. In this context, the role of bilirubin in protecting from the development of liver disease, by activating the PPARα that is ultimately responsible for fatty acid transport and peroxisomal and mitochondrial fatty acid β-oxidation has been proposed [18,32,57,60]. Therefore, the association found between VAT BVR-A mRNA levels and the histological diagnosis and severity of NAFLD observed in our study, further suggest that additional mechanisms likely involving BVR-A may have a role. Indeed, VAT expands during obesity and secrets inflammatory factors that contribute to low-grade chronic inflammation also responsible for NAFLD development [15]. Furthermore, consistent with the hypothesis about a role for BVR-A, we previously reported that in PBMC collected from obese and lean individuals, reduced BVR-A levels and activation were associated with increased NAFLD and VAT inflammation [34]. Hence, the finding that lower BVR-A mRNA expression levels are significantly associated with a higher prevalence of NAFLD appears of interest and clinically relevant.

We acknowledge that the design of our study does not allow us to establish a causal nexus between lower VAT BVR-A mRNA expression and increased VAT dysfunction and NAFLD development and thus further studies are warranted to fully understand the pathophysiologic processes behind our observations. In particular, a limitation of our study is represented by the lack of comparison with measures performed in VAT samples from normal-weight individuals. However, collecting VAT samples from normal-weight individuals raise ethical concerns. Moreover, for this study, BVR-A expression levels were only measured in VAT and not in subcutaneous fat; besides, data on differential VAT and SAT BVR-A expression may provide insights on mechanisms of insulin resistance in obesity and warrant to be investigated on specifically designed studies.

## 4. Materials and Methods

### 4.1. Study Population

For this study, we recruited 38 obese individual candidates for bariatric surgery at Sapienza University of Rome, Italy. Patients with body mass index (BMI) ≥35 kg/m^2^ and without a history of excessive alcohol consumption or other causes of hepatic diseases were eligible for this study; the inclusion and exclusion criteria met those reported in the International guidelines for sleeve gastrectomy [61]. For all study participants, we collected medical history and evaluated anthropometric parameters including height, weight, waist circumference, and BMI calculation. Systemic systolic (SBP) and diastolic (DBP) blood pressure were recorded as an average of three measurements assessed after 5 min rest. MS was defined according to the modified NCEP ATP-III criteria [62] and diabetes mellitus was diagnosed based on the American Diabetes Association 2020 criteria [63].

### 4.2. Laboratory Measurements

In all study participants, routine biochemistry was performed on overnight fasting blood samples. Fasting blood glucose (FBG, mg/dL), glycosylated haemoglobin (HbA1c, %—mmol/mol), total cholesterol (mg/dL), high-density lipoprotein cholesterol (HDL, mg/dL), triglycerides (mg/dL), aspartate aminotransferase (AST, IU/L), alanine aminotransferase (ALT, IU/L), gamma-glutamyl transpeptidase (GGT, IU/L), and creatinine (mg/dL) were measured by standard laboratory methods. The low-density lipoprotein (LDL) cholesterol value was calculated by the Friedwald formula.

### 4.3. Histological and Gene Expression Analyses

Histological examinations were performed on paired omental and hepatic biopsies obtained intra-operatively during laparoscopic sleeve gastrectomy of obese individuals.

The storage and the analysis of VAT biopsies were performed as previously described [64]. A computerized digital camera (Olympus Camedia 5050, Olympus Inc., Tokyo, Japan) was used to capture the images (×200 magnification) of VAT (stored as JPG files), and analyzed for morphometric evaluation of adipocyte cross-sectional area (CSA), with computerized imaging software (ImageJ 1.53c, NIH, Bethesda, Maryland). CSA was evaluated in 400 adipose cells transversally sectioned. The diagnosis of NAFLD/NASH was performed on liver biopsies based on validated criteria [65,66]. Experimental procedures for liver fragments examination are detailed in our previous study [67]. Two pathologists blinded to patients’ identity performed all the histological analyses.

Human BVR-A (gene ID: 644), IL-8 (gene ID: 624), Caspase 3 (gene ID: 836), IL-6 (gene ID: 3569), and TNFα (gene ID: 7124) mRNA were detected by using real-time quantitative PCR performed as detailed in our previous study [68].

### 4.4. Statistical Analysis

All the statistical analyses were performed using SPSS version 25 (IBM, Armonk, NY). Variables are reported as mean ± standard deviation (SD), median (interquartile range) value, or percentages, as appropriate. Comparisons between groups were performed by Student’s t-test for normally distributed variables, Mann–Whitney for non-normally distributed variables, or χ^2^ test for categorical parameters. Pearson and Spearman’s rank correlation coefficients were used to evaluate bivariate correlations. The predictive value of BVR-A expression for NAFLD diagnosis (yes/no) was estimated by calculating the receiver-operating characteristic (ROC) curve adjusted for age, sex, and BMI with a 95% confidence interval (C.I.). To the best of our knowledge, no study has so far investigated the VAT BVR-A expression in relation to VAT inflammation and metabolic impairment in humans. Therefore, the statistical power of this study was confirmed by a post-hoc sample size calculation considering the mean difference of IL-8 expression levels between subjects belonging to the low and high BVR-A group; thus, we obtained that 26 patients were enough to reach statistical significance with power = 80% and *α* error = 0.05. Two-sided *p*-value < 0.05 was considered statistically significant, with a confidence interval of 95%.

### 4.5. Ethics Standards

The study protocol has been reviewed and approved by the Ethics Committee of Policlinico Umberto I (approval number 3550, 26 February 2015); the study was conducted in conformance with the Helsinki Declaration. All the study participants signed written informed consent before undergoing all the study procedures.

## 5. Conclusions

Our study provides, for the first time in humans, insights into the association between VAT BVR-A mRNA levels and markers of VAT dysfunction and NAFLD. We propose that BVR-A is engaged in regulating VAT homeostasis and its impairment could be involved in the dysfunctional remodeling of AT occurring in obesity. These observations are consistent with data available in the literature obtained from both in vitro and in vivo experimental models and extend our knowledge about the anti-inflammatory role of BVR-A in metabolic disorders such as obesity.

## Figures and Tables

**Figure 1 ijms-21-09091-f001:**
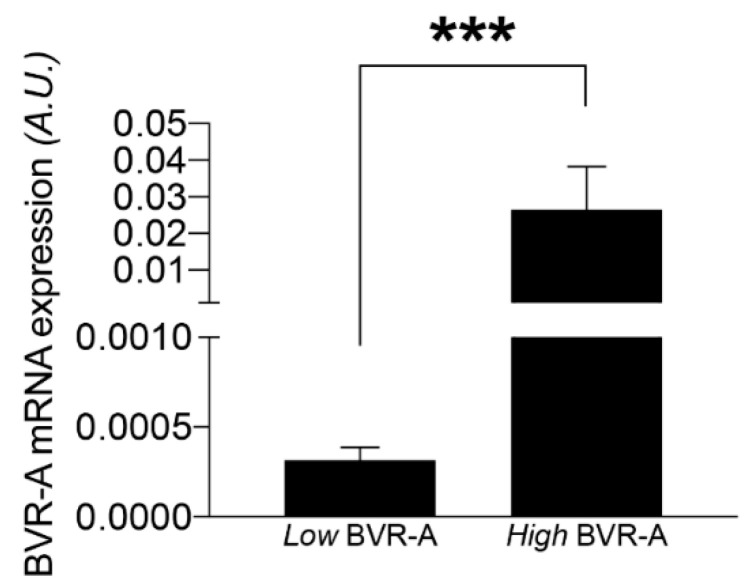
mRNA expression levels of BVR-A in VAT. Subjects were dived into two groups based on BVR-A mRNA expression levels below or above the median value (low and high BVR-A, respectively). Data are expressed as mean ± SEM, *** *p* < 0.0001 (Mann–Whitney). A.U., arbitrary units.

**Figure 2 ijms-21-09091-f002:**
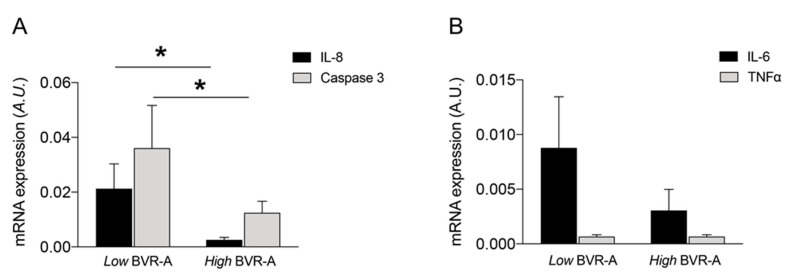
IL-8, Caspase 3 (**A**), IL-6 and TNF-α (**B**) mRNA expression levels in VAT relatively to BVR-A mRNA expression levels. Subjects were dived into two groups based on BVR-A mRNA expression levels below or above the median value (low and high BVR-A, respectively). Data are expressed as mean ± SEM, * *p* < 0.05 (Mann–Whitney). A.U., arbitrary units.

**Figure 3 ijms-21-09091-f003:**
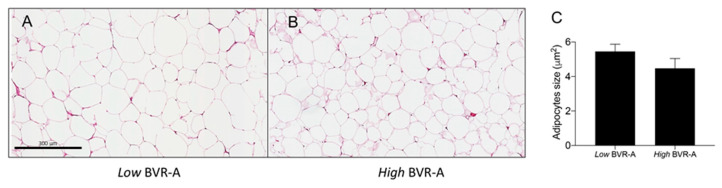
Images showing adipocytes of two representative study participants belonging to low BVR-A group (**A**) and high BVR-A group (**B**). In (**C**) quantification of the cross-sectional area (CSA) expressed as mean ± SEM.

**Figure 4 ijms-21-09091-f004:**
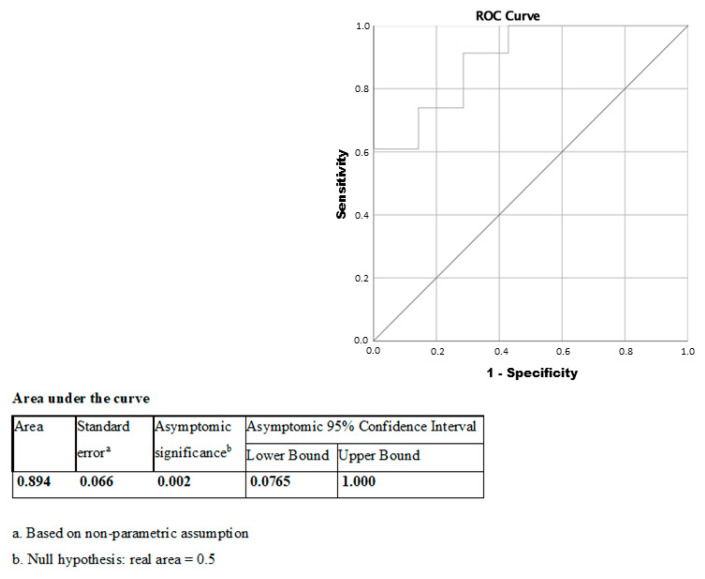
VAT BVR-A mRNA expression receiver operating curve (ROC) for NAFLD.

**Table 1 ijms-21-09091-t001:** Characteristics of the study population in relation to the BVR-A mRNA expression levels in the VAT.

	Low BVR-A (*n* = 19)	High BVR-A (*n* = 19)	*p*-Value
Age (years)	45.4 ± 10	42.4 ± 9.3	0.42
Gender (F%)	73%	73%	0.94
BMI (Kg/m^2^)	43 ± 6.4	42.5 ± 3.96	0.92
Waist circumference (cm)	130.4 ± 9	126.9 ± 11.4	0.44
SBP (mmHg)	127.5 ± 9.76	125 ± 12.7	0.37
DBP (mmHg)	88.2 ± 29.1	83.2 ± 9.1	0.68
FBG (mg/dL)	103.9 ± 16.7	96.6 ± 14.5	0.16
HbA1c (%—mmol/mol)	5.5 ± 0.3	5.4 ± 0.5	0.35
FBI (µU/L)	12.7 ± 7.2	14.1 ± 7.6	0.67
HOMA-IR	3.3 ± 2	3.2 ± 1.8	1.0
HOMA-β %	122.2 ± 67.1	182.1 ± 141.5	0.22
Total Cholesterol (mg/dL)	206.5 ± 34.5	196.3 ± 25.6	0.52
HDL (mg/dL)	51.9 ± 11.8	45.6 ± 7.4	0.18
LDL (mg/dL)	128 ± 28	124.3 ± 24.6	0.52
Triglycerides (mg/dL)	131.2 ± 40.4	120.7 ± 47.3	0.50
AST(IU/L)	23.9 ± 10.2	21.7 ± 9	0.59
ALT (IU/L)	30.2 ± 17.2	27.1 ± 13.6	0.88
GGT (IU/L)	36 ± 43.2	19.5 ± 7.7	0.05
Total Bilirubin (mg/dl)	0.68 (0.5–0.98)	0.71 (0.6–1.02)	0.24
Conjugated Bilirubin (mg/dl)	0.36 (0.19–0.39)	0.27 (0.16–0.43)	0.80
Serum Creatinine (mg/dL)	0.79 ± 0.2	0.79 ± 0.1	0.91
Uric Acid (mg/dL)	5.8 ± 1.7	5.5 ± 1.2	0.84
T2D (%)	8%	13%	0.63
MS (%)	82%	88%	0.58
NAFLD (%)	95%	68%	0.036

(Low BVR-A: subgroup with VAT BVR-A below median BVR-A levels; high BVR-A: subgroup with VAT BVR-A above median BVR-A levels). Values are expressed by mean ± SD or percentage, as appropriate. Mann–Whitney test, χ^2^ test applied.

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
