# Peer review of "Reduced Biliverdin Reductase-A Expression in Visceral Adipose Tissue is Associated with Adipocyte Dysfunction and NAFLD in Human Obesity"

_ijms, 2020, doi:10.3390/ijms21239091_

Round 1
Reviewer 1 Report
The paper by Ceccarelli et al. entitled ¨Reduced biliverdin reductase-A expression in visceral adipose tissue is associated with adipocyte dysfunction and NAFLD in human obesity¨ deals with an important topic of the role of BVRA in the pathogenesis of obesity and NAFLD. This is an interesting clinical study, which needs, however, clarification of certain aspects.
Major comments:
- NAFLD diagnosis is based on liver histology, which is certainly great, but authors do not comment on NAFLD grading, or presence of NASH (non-alcoholic steatohepatitis), which, provided that the same trend of BVRA will be observed, would further enhance the importance of BVRA pathway for its pathogenesis. It is likely that the patients had also ultrasonographic examination of the liver, and perhaps also elastography, the authors should also add this data a perform the correlations.
- BVRA mRNA expressions were performed in the omental adipose tissue. Why did not the authors analyze also liver expression, when liver tissue was available. Could they add this analyses?
- Bilirubin concentrations in both cohorts are rather high, much higher than what is common in NAFLD/NASH. There is an abundance of clinical reports demonstrating much lower serum bilirubin concentrations in NAFLD/NASH patients. On contrary, NASH is rare in subjects with benign hyperbilirubinemia above 1 mg/dL (see Hjelkrem APT 2012), which was certainly relatively frequently seen in the current study, despite the fact, that liver enzyme activities were surprisingly low – which also need more detailed explanation. Elevation of liver enzyme activities (ALT/AST/GGT) is seen in 10-20% of general population, and in much higher proportions in extremely obese, NAFLD patients with GGT being a secondary marker of metabolic syndrome. It is thus very surprising to see such low values in the studied patients, and authors should try to explain this observation.
- Serum bilirubin concentrations should be given as median and IQ range, since they do not have normal distribution in population.
- When discussing liver enzymes, the term ¨levels¨ should be exchanged for ¨activities¨.
Minor comment
¨glutamil¨ should read ¨glutamyl¨
Author Response
- NAFLD diagnosis is based on liver histology, which is certainly great, but authors do not comment on NAFLD grading, or presence of NASH (non-alcoholic steatohepatitis), which, provided that the same trend of BVRA will be observed, would further enhance the importance of BVRA pathway for its pathogenesis. It is likely that the patients had also ultrasonographic examination of the liver, and perhaps also elastography, the authors should also add this data a perform the correlations.
We agree with the reviewer that data on a linear association between low BVR-A expression and worse NAFLD grading would strengthen the significance of our results. In order to address this important point, we performed new statistical analyses showing that the subgroup of individuals without detectable BVR-A expression in the VAT (n= 6) had significantly more severe liver steatosis, as considered by greater Brunt steatosis score, than those with higher VAT BVR-A levels (p=0.03). This finding provides further support to the hypothesis of an involvement of impaired BVR-A signalling in the development of liver insulin resistance and intrahepatic fat accumulation in obesity. Although liver ultrasonography is the most common imaging technique to screen individuals for fatty liver disease, in presence of severe obesity, the sensitivity of this diagnostic tool drops to less than 40%, due to technical difficulties in its execution (Wieckowska A et al. 2007). Therefore, for the purposes of this study, we only considered data from liver histology, which is largely considered as the gold standard technique for NAFLD and NASH diagnosis (Kleiner DE et al., 2005; Kleiner DE et al., 2012). We thank the reviewer for this comment and have now added these new analyses in the Results section (page 4, lines 112-114).
- BVRA mRNA expressions were performed in the omental adipose tissue. Why did not the authors analyze also liver expression, when liver tissue was available. Could they add this analyses?
In the present study, starting from initial experimental evidence, we aimed to explore BVR-A expression in the adipose tissue and its metabolic correlates in humans. However, as pointed out by this reviewer, data on hepatic BVR-A levels are of sure interest in order to better characterize BVR-A pathways in relation to insulin resistance and associated disorders. Indeed, in parallel to the BVR-A expression study in AT, our research group is performing also investigations on BVR-A expression levels in liver from obese subjects, and experimental procedures are currently ongoing. In the preliminary statistical analysis, hepatic BVR-A expression was significantly associated with the presence of liver damage, as expressed by greater Brunt (p= 0.04) and Nas Score (p= 0.03). Thus, in obesity, impaired hepatic fatty acids beta-oxidation and bile acids pool might trigger the local expression of BVR-A in the liver, as a compensatory mechanism to metabolize the excess of substrates. Further analyses are warranted in order to confirm these initial findings.
- Bilirubin concentrations in both cohorts are rather high, much higher than what is common in NAFLD/NASH. There is an abundance of clinical reports demonstrating much lower serum bilirubin concentrations in NAFLD/NASH patients. On contrary, NASH is rare in subjects with benign hyperbilirubinemia above 1 mg/dL (see Hjelkrem APT 2012), which was certainly relatively frequently seen in the current study, despite the fact, that liver enzyme activities were surprisingly low – which also need more detailed explanation. Elevation of liver enzyme activities (ALT/AST/GGT) is seen in 10-20% of general population, and in much higher proportions in extremely obese, NAFLD patients with GGT being a secondary marker of metabolic syndrome. It is thus very surprising to see such low values in the studied patients, and authors should try to explain this observation.
We agree with the reviewer that, when considering mean and standard deviation values, total and conjugated bilirubin concentrations in our study population seemed higher than what is commonly reported in NAFLD/NASH. Indeed, as also pointed out by this reviewer in Comment #4, bilirubin has a skewed distribution and therefore mean and standard deviation does not describe appropriately its distribution in our study sample. Therefore, we have now calculated median and IQ range for total and conjugated bilirubin in relation to the presence of NAFLD and NASH, finding values within the normal range of our lab in all the three subgroups [median (IQ) total and conjugated bilirubin in no-NAFLD group: 0.55(0.37-0.70) and 0.16 (0.12-0.23) mg/dl, respectively; NAFLD group: 0.76 (0.62-0.97) and 0.34 (0.16-0.44) mg/dl; NASH group: 0.70 (0.43-0.76) mg/dl and conjugated bilirubin: 0.22 (0.14-0.36) mg/dl]. Differences of total and conjugated bilirubin among no-NAFL, NAFLD and NASH individuals were not statistically significant. We apologize if our data report was misleading and have now corrected it in the manuscript (Table 1, see also response to Comment #4).
Regarding serum transaminases activities, in our population of obese individuals, mean AST, ALT and GGT levels fell within the normal range. However, when stratifying in relation to the presence of biopsy-proven NAFLD/NASH, we found that NAFLD/NASH patients had significantly higher ALT and GGT activities than obese individuals without NAFLD, as expected (ALT: 32.9 ± 15.8 vs 17.2 ± 4 IU/L, p= 0.001; GGT: 30.7 ± 34.6 vs 17.3 ± 4.2, p= 0.05). Another important point is that study population had rather low mean age and prevalence of advanced metabolic disease, such as T2D, detected in less than 10% of study participants. Moreover, no individual had advanced hepatic damage at the liver histology (fibrosis score (F)/number of participants: (F0/F1/F2/F3/F4: 8/26/3/1/0), thus providing further explanation to the low transaminases values found in the studied patients.
- Serum bilirubin concentrations should be given as median and IQ range, since they do not have normal distribution in population.
Serum total and conjugated bilirubin concentrations are now given as median and IQ range in Table 1 (Page 3).
- When discussing liver enzymes, the term ¨levels¨ should be exchanged for ¨activities¨.
The term ¨levels¨ has been substituted with the term ¨activities” in the manuscript, as suggested by the reviewer (page 3, line 81; page 4, line 110).
- Minor comment: ¨glutamil¨ should read ¨glutamyl¨
This mistake has been corrected (Page 1, line 33).
Reviewer 2 Report
The article presented by Ceccarelli et al., titled Reduced biliverdin reductase-A expression in visceral adipose tissue is associated with adipocyte dysfunction and NAFLD in human obesity is an interesting study that highlights an important concept regarding a relatively novel phenomenon. The use of a human tissue samples is important for capturing a snapshot of the inflammatory transcriptome and histological differences in adipocyte tissue morphology in obese patients.
While an interesting article, it is important that the authors address a few concepts to bring awareness and understanding to the specific field they are exploring.
- Is it important to consider and address the role of Subcutaneous Adipose Tissue (SAT) and its ability to function (dis)synergistically with VAT? For example, could SAT with elevated BVR-A contribute to the changes seen in the VAT (via unconjugated/conjugated bilirubin, adipokines, etc?)
- Is the difference between conjugated (direct) vs unconjugated (indirect) levels of bilirubin important? There seems to be little to no importance of conjugated bilirubin, does your study suggest otherwise?
- Recent studies have expounded upon the direct role bilirubin/BVR-A has on lipid regulation with PPAR alpha (the lipolytic isoform) vs the other PPAR isoforms. An article was recently published (PMID: 32404366) describing the molecular framework of bilirubin’s mediated changes in lipid regulation via both lipolytic and thermogenic pathways. In addition, the synergistic nature of BVR-A in increasing stabilizaiton of PPARa and bilirubin’s direct activation of PPARa (PMID: 32404366) may be an important phenomenon to discuss. Has there been a measurement of the lipolytic/thermogenic genes from the tissue samples?
Author Response
- Is it important to consider and address the role of Subcutaneous Adipose Tissue (SAT) and its ability to function (dis)synergistically with VAT? For example, could SAT with elevated BVR-A contribute to the changes seen in the VAT (via unconjugated/conjugated bilirubin, adipokines, etc?)
We thank the reviewer for this important question, which we are currently trying to address in newly designed studies. Indeed, we recently started a novel project aimed to explore differential BVR-A and inflammatory patterns in subcutaneous vs visceral adipose tissue. Our hypothesis is that BVR-A may have a role in the regulation of insulin signaling and associated chronic low-grade inflammation and this modulation may differ in relation of AT site. Collection of both VAT and SAT samples from subjects undergoing bariatric surgery is ongoing. Conversely, as for this study design, we were only able to obtain samples from omental tissue collected intra-operatively during sleeve gastrectomy procedure. We have now acknowledged this point as a study limitation in the Discussion section of this manuscript: “Moreover, for the purposes of this study, BVR-A expression levels were only measured in VAT and not in subcutaneous fat; besides, data on differential VAT and SAT BVR-A expression may provide insights on mechanisms of insulin resistance in obesity and warrant to be investigated in specifically designed studies” (page 7, lines 203-207).
- Is the difference between conjugated (direct) vs unconjugated (indirect) levels of bilirubin important? There seems to be little to no importance of conjugated bilirubin, does your study suggest otherwise?
Our study did not shown any difference between conjugated and unconjugated bilirubin levels in relation to VAT BVR-A expression and did not support a specific importance of conjugated bilirubin in obesity-associated metabolic disorders in our study population. Indeed, we have calculated median (IQR) total and conjugated bilirubin levels in relation to the presence of liver damage, showing that they did not differ among no-NAFL, NAFLD and NASH individuals [median (IQ) total and conjugated bilirubin in no-NAFLD group: 0.55(0.37-0.70) and 0.16 (0.12-0.23) mg/dl, respectively; NAFLD group: 0.76 (0.62-0.97) and 0.34 (0.16-0.44) mg/dl; NASH group: 0.70 (0.43-0.76) mg/dl and conjugated bilirubin: 0.22 (0.14-0.36) mg/dl, p= not significant for all the above; see also response to Comment #3, Reviewer 1].
- Recent studies have expounded upon the direct role bilirubin/BVR-A has on lipid regulation with PPAR alpha (the lipolytic isoform) vs the other PPAR isoforms. An article was recently published (PMID: 32404366) describing the molecular framework of bilirubin’s mediated changes in lipid regulation via both lipolytic and thermogenic pathways. In addition, the synergistic nature of BVR-A in increasing stabilization of PPARa and bilirubin’s direct activation of PPARa (PMID: 32404366) may be an important phenomenon to discuss. Has there been a measurement of the lipolytic/thermogenic genes from the tissue samples?
We thank the reviewer also for this valuable comment. Increased bilirubin serum levels were shown to decrease body weight and to reduce lipid accumulation in experimental models (Stec DE et al, 2016; Takei R et al., 2019). These events seem to be mediated by a direct interaction between bilirubin and PPARa, which results in the activation of lipid-burning genes such as UCP1 and CPT1, particularly in the VAT adipocytes (Gordon DM et al., 2020). In addition, bilirubin-mediated PPARa-induced activation of UCP1 and CPT1 also results in an enhancement of mitochondrial functions, finally responsible for lipid burning and weight loss (Gordon DM et al., 2020). Together, these findings suggest that bilirubin may regulate VAT tissue expansion, reduce VAT hypertrophy and improve glucose metabolism. Our results about reduced adipocytes size in the high-BVR-A group is in line with these experimental observations. Indeed, higher BVR-A levels could favor an increased bilirubin production within the adipocytes thus promoting lipid-burning effects. In support of this hypothesis, Stec et al. reported that BVR-A knock-out in adipocytes promotes a reduction of mitochondrial density, further impairing the lipid-burning effect (Stec DE et al., 2020). We have now added a paragraph discussing this aspect, along with related references, in the revised version of our manuscript: “Increased bilirubin serum levels were shown to decrease body weight and to reduce lipid accumulation in experimental models [57-58], likely by a direct interaction between bilirubin and PPARa, resulting in the activation of lipid-burning genes -such as UCP1 and CPT1- in VAT adipocytes [59]. Bilirubin-mediated PPARa-induced activation of UCP1 and CPT1 also results in enhanced mitochondrial function, finally promoting lipid burning and weight loss [59]. Altogether these findings suggest that bilirubin may regulate VAT tissue expansion, reduce VAT hypertrophy and improve glucose metabolism. Our results on smaller adipocytes size in the high-BVR-A group are in line with these experimental observations. Indeed, higher BVR-A levels could favor an increased bilirubin production within the adipocytes thus promoting lipid-burning effects” (page 6, lines 168-175).
Round 2
Reviewer 1 Report
The authors addressed all the issues, I do not hyve further comments.